# Axonal Protection by Oral Nicotinamide Riboside Treatment with Upregulated AMPK Phosphorylation in a Rat Glaucomatous Degeneration Model

**Ibuki Arizono [1,2], Naoki Fujita [2], Chihiro Tsukahara [2], Kana Sase [2], Reio Sekine [2], Tatsuya Jujo [2], Mizuki Otsubo [1,2], Naoto Tokuda [2] and Yasushi Kitaoka [1,2,*]**

[1] Department of Molecular Neuroscience, St. Marianna University Graduate School of Medicine, Kawasaki 216-8511, Japan; ibuki.arizono@marianna-u.ac.jp (I.A.)

[2] Department of Ophthalmology, St. Marianna University School of Medicine, Kawasaki 216-8511, Japan

[*] Correspondence: kitaoka@marianna-u.ac.jp

**Abstract:** Nicotinamide riboside (NR), a precursor of nicotinamide adenine dinucleotide (NAD$^+$), has been studied to support human health against metabolic stress, cardiovascular disease, and neurodegenerative disease. In the present study, we investigated the effects of oral NR on axonal damage in a rat ocular hypertension model. Intraocular pressure (IOP) elevation was induced by laser irradiation and then the rats received oral NR of 1000 mg/kg/day daily. IOP elevation was seen 7, 14, and 21 days after laser irradiation compared with the controls. We confirmed that oral NR administration significantly increased NAD$^+$ levels in the retina. After 3-week oral administration of NR, morphometric analysis of optic nerve cross-sections showed that the number of axons was protected compared with that in the untreated ocular hypertension group. Oral NR administration significantly prevented retinal ganglion cell (RGC) fiber loss in retinal flat mounts, as shown by neurofilament immunostaining. Immunoblotting samples from the optic nerves showed that oral NR administration augmented the phosphorylated adenosine monophosphate-activated protein kinase (p-AMPK) level in rats with and without ocular hypertension induction. Immunohistochemical analysis showed that some p-AMPK-immunopositive fibers were colocalized with neurofilament immunoreactivity in the control group, and oral NR administration enhanced p-AMPK immunopositivity. Our findings suggest that oral NR administration protects against glaucomatous RGC axonal degeneration with the possible upregulation of p-AMPK.

**Keywords:** nicotinamide riboside; p-AMPK; glaucoma; optic nerve; RGC

## 1. Introduction

Nicotinamide riboside (NR) is available as a nicotinamide adenine dinucleotide$^+$ (NAD$^+$) precursor supplement and was reported to upregulate serum NAD$^+$ levels in humans [1]. A recent study reported a significantly lower plasma nicotinamide concentration in primary open-angle glaucoma patients compared with the control group [2]. Because the oral intake of vitamin B3/nicotinamide augments NAD levels in the retina and exerts axonal protection in the spontaneous glaucomatous degeneration DBA/2J mouse model [3], and oral intake of nicotinamide provides neuroprotection in a rat bead model of ocular hypertension [4], it is reasonable to speculate that nicotinamide supplementation may have favorable effects in certain types of glaucomatous damage. A recent clinical study showed that oral nicotinamide improved retinal ganglion cell (RGC) function as assessed by an electroretinogram in glaucoma patients [5]. In addition, a more recent clinical study in glaucoma patients demonstrated that the oral intake of nicotinamide and pyruvate yielded improvement of pattern standard deviation in the treatment group compared with the placebo group [6]. Because several neuroprotective potential treatments have been discussed in glaucoma [7] and the health of RGC is heavily dependent upon NAD$^+$ [8], it is

important to elucidate the molecular mechanisms of action of nicotinamide on RGC axons in glaucomatous damage.

A previous large clinical study showed that metformin is associated with a reduced risk of developing open-angle glaucoma in diabetes mellitus patients [9]. Several groups demonstrated that metformin activates the adenosine monophosphate-activated protein kinase (AMPK) signaling pathway in neuronal systems [10–12], and many studies suggested beneficial effects of AMPK in certain neuronal cells [13,14]. For example, 5-aminoimidazole-4-carboxamide ribonucleotide (AICAR), an AMPK activator, improved depressive behavior and enhanced hippocampal neurogenesis in olfactory-injured mice [15]. In addition, A769662, another AMPK activator, accelerated clearance of $\alpha$-synuclein aggregates in SH-SY5Y cells [16]. It was shown that the AMPK activator AICAR protected photoreceptors against ocular light-exposure injury [17]. Although some recent studies suggested a close relationship between nicotinamide and the AMPK pathway in non-neuronal cells [18,19], and a recent systematic review suggested that many $NAD^+$ regulators affect the AMPK pathway with the potential for the treatment of various diseases [20], this relationship was not examined in RGC axons. Thus, the purpose of the present study was to examine the effects of oral NR administration on glaucomatous axonal degeneration and to investigate whether it alters phosphorylated AMPK (p-AMPK) expression in the optic nerve.

## 2. Material and Methods

### 2.1. Animals

Experiments were carried out on 8-week-old male Wistar rats. All studies were conducted according to the ARVO Statement for the Use of Animals in Ophthalmic and Vision Research and approved by the Ethics Committee of the Institute of Experimental Animals, St. Marianna University School of Medicine (No. 2108012, No. 2208007). The animals were housed in a controlled environment ($23 \pm 1$ °C; humidity $55 \pm 5\%$; light from 6 a.m. to 6 p.m.) throughout the experimental period.

### 2.2. Ocular Hypertension Model

A rat ocular hypertension model was generated as described previously [21]. Briefly, rats were anesthetized with intramuscular injections of a mixture of ketamine–xylazine, and 10 µL of India ink (Remel Europe, Ltd., Dartford, UK) was injected intracamerally to the right eye using a Hamilton syringe with a 30-gauge needle. Laser irradiation was delivered ab externo to the pigmented trabecular band, which appears as a dark circumferential area at an argon laser setting of 200 µm spot, 200 mW, 200 ms, totaling 180–200 shots (Novus Varia, Lumenis, Dieburg, Germany) 3 days after ink injection. The contralateral left eye without India ink was used as the control or NR alone group. For the sham study, we injected India ink intracamerally to the right eye and compared the IOP between the India ink group and India ink with the laser irradiation group.

### 2.3. Drug Administration

NR chloride (NIAGEN; ChromaDex, Inc. Los Angeles, CA, USA) 1000 mg/kg dissolved in 0.5% methylcellulose in distilled water was administered orally via an oral Zonde needle immediately before laser irradiation in conscious rats. Subsequently, NR was administered orally once daily for 3 weeks. One and three weeks after laser irradiation, the rats were euthanatized with an overdose of sodium pentobarbital and the eyes were enucleated. In addition, we examined p-AMPK level in the optic nerve a short time after one-time NR administration. For this experiment, the eyes were enucleated 6, 24, and 48 hrs after NR administration.

### 2.4. IOP Measurement

IOP was measured prior to ink injection (baseline) and 1, 2, and 3 weeks after laser irradiation. For the sham study, IOP was measured at baseline and 1, 3, and 5 days after laser irradiation. Rat IOP was measured while conscious using a portable tonometer

(Tonolab, Icare Finland, Helsinki, Finland) at almost the same time in the evening. Six consecutive readings were performed, and the first successful measurement was used as the result for analysis after eliminating the minimum and maximum measurements.

### 2.5. Immunoblotting

To examine the effect of NR on p-AMPK in the glaucomatous optic nerve, optic nerve specimens (4 mm proximal lengths) were homogenized in protein extraction buffer 1 week after laser irradiation. To examine the effect of NR on p-AMPK at short time periods, optic nerve samples were collected 6, 24, and 48 hrs after one-time NR administration. One optic nerve preparation included two optic nerves. Homogenized samples were centrifuged at $15,000 \times g$ for 15 min at 4 °C. Protein concentrations were determined with the supernatants. Each sample (3 μg) was subjected to the mini gel (Bio-Rad Laboratories, Hercules, CA, USA) and transferred to an enhanced chemiluminescent membrane (EMD Millipore Corporation, Temecula, CA, USA). The membranes were blocked with 5% skim milk with tris-buffered saline (TBS) containing Tween-20 and reacted with anti-p-AMPK antibody (1:200; Sigma-Aldrich, St. Louis, MO, USA), anti-AMPK antibody (1:200; Sigma-Aldrich), or anti-β-actin antibody (1:5000; Sigma-Aldrich). After washing three times, the membranes were reacted with anti-rabbit or anti-mouse peroxidase-labeled secondary antibody (1:5000; MP Biochemicals, Solo, OH, USA). Immunoblotting was visualized with a chemiluminescence detection system (ECL Plus Western Blotting Detection Re000000agents, Amersham Pharmacia Biotech, Buckinghamshire, UK).

### 2.6. Immunohistochemistry

Three eyes 1 week after laser irradiation or three normal eyes were collected and fixed by immersion in 4% paraformaldehyde, dehydrated, and embedded in paraffin. Sections (4 μm thickness) were made through the optic disc and blocked with 1% bovine serum (Roche Diagnostics GmbH, Mannheim, Germany). The primary antibodies were against p-AMPK (1:100; Sigma-Aldrich) and neurofilament-L (a marker of nerve fibers; 1:100; Dako, Tokyo, Japan). The secondary antibodies were FITC-labeled or rhodamine-labeled antibodies (1:100; Cappel, Aurora, OH, USA). The sections were mounted on slides in a DAPI-containing medium under the cover of glass. The images were captured using a confocal microscopy system (Zen; Carl Zeiss QEC GmbH, Köln, Germany).

### 2.7. Quantification of Optic Nerve Axons

Three weeks after laser irradiation, optic nerve specimens (4 mm lengths from 1 mm behind the globe) were collected and soaked in Karnovsky's solution for 24 h at 4 °C. Several dehydrations were performed, and samples were embedded in acrylic resin at 70 °C for 48 h. Then, the samples were sectioned and stained with 1% paraphenylene-diamine (Sigma-Aldrich) in absolute methanol [22,23] to stain myelin. Five black and white images from each eye were obtained at the center and at each quadrant of the periphery with a light microscope (Olympus, Tokyo, Japan). These black and white images (each area, 5850 μm$^2$; total area, 29,250 μm$^2$ per eye) were used for quantification with the Aphelion image processing software (version 3.2. ADCIS S.A., Hérouville Saint-Clair, Caen, France). The number of axons was averaged in each eye and each group, and data are presented as the number per square millimeter. After quantification, representative color photos were obtained.

### 2.8. Retinal Flat-Mount Immunostaining of Neurofilament

Three weeks after laser irradiation, eyes were soaked in 4% paraformaldehyde at 4 °C for 2 h. Retinal cups were carefully separated from the eyes and re-soaked in 4% paraformaldehyde for 2 h at 4 °C. After five washes with PBS for 5 min each, the cups were incubated for 1 h in blocking buffer (0.5% triton and 1% BSA in PBS) at 4 °C. The cups were then incubated with the primary antibody, mouse neurofilament-L (a marker of nerve fibers; 1:100; Dako), overnight at 4 °C. After three washes with blocking buffer

for 20 min each at 4 °C, the cups were incubated for 6 h in the corresponding secondary antibody, FITC-labeled anti-mouse antibody (1:100; Cappel), at 4 °C. After 4 washes with PBS for 15 min each, these retinal samples were flat-mounted and fluorescence microscope (Olympus, Tokyo, Japan) images were taken. The flat-mounted retinal images were then analyzed using Q-Capture Pro 7 (QImaging, British Columbia, Canada). Central (500 μm from the edge of the optic disc) and peripheral (2 mm from the edge of the optic disc) quadrant images, totaling 8 images, were taken. The analysis line length was 500 μm, including 1450 pixels. The central and peripheral quadrant analysis lines were used for pixel brightness calculations. Five points on the line were used to determine the background value in one image, and this background value was subtracted from each pixel brightness result. The pixel brightness values were expressed as the area under the curve, which means the total sum of the pixel value in the analysis line length (Supplementary Figure S1). Data from 4 central and 4 peripheral fields in each eye were averaged, and 4 control eyes, 6 ocular hypertensive eyes, and 6 ocular hypertensive eyes after NR administration were used to calculate the results.

### 2.9. Determination of Retinal $NAD^+$ Levels

The retinal $NAD^+$ levels were determined using the $NAD^+$/NADH Assay Kit (Dojindo, Kumamoto, Japan) following the manufacturer's instructions. After 1 week of oral NR administration, retinas (one retinal sample consisted of 4 retinas) were collected and homogenized in an ice-cold buffer containing 0.25 M potassium hydroxide. A total of 7 samples (28 retinas) were included in both the control and NR groups (56 retinas). The homogenate was centrifuged at $12,000\times g$ for 10 min at 4 °C, and then the supernatant was transferred to a 10 KDa filtration tube and centrifuged at $15,000\times g$ for 90 min at 4 °C. To measure retinal NADH levels, the same samples were incubated at 60 °C for 60 min to degrade retinal NAD. All samples were then neutralized with 1 mol/l $KH_2PO_4$ solution and then loaded onto microtiter plates, mixed with the NADH developer, and incubated at room temperature for 1 h. The absorbance of each sample was measured at the wavelength of 450 nm. $NAD^+$ levels were calculated by subtracting the NADH level from the NAD total.

### 2.10. Statistical Analysis

Data are expressed as mean $\pm$ SEM. Differences among groups were analyzed using one-way ANOVA with the post hoc Tukey's HSD test or Student's *t*-test method. A *p* value of <0.05 was considered to represent a statistically significant difference. All graphs were drawn using GraphPad Prism 9 software (GraphPad Software, Boston, MA, USA).

## 3. Results

### 3.1. IOP Measurement

There was no significant difference in IOP at baseline among the groups (Figure 1). Significant IOP elevation was seen at 7, 14, and 21 days both with and without NR administration after laser irradiation compared with the control (Figure 1). These IOP elevations were greater than 30 mmHg, and the values are consistent with our previous results [21]. There was no significant difference between the ocular hypertension group and the ocular hypertension + NR administration group. Moreover, there was no significant difference between the control and NR groups. In the sham study, there was no significant difference in IOP at baseline between the two groups (Supplementary Figure S2). Although IOP tended to be higher in the laser group compared to the no-laser group, statistical significance was observed on only day 1 (Supplementary Figure S2).

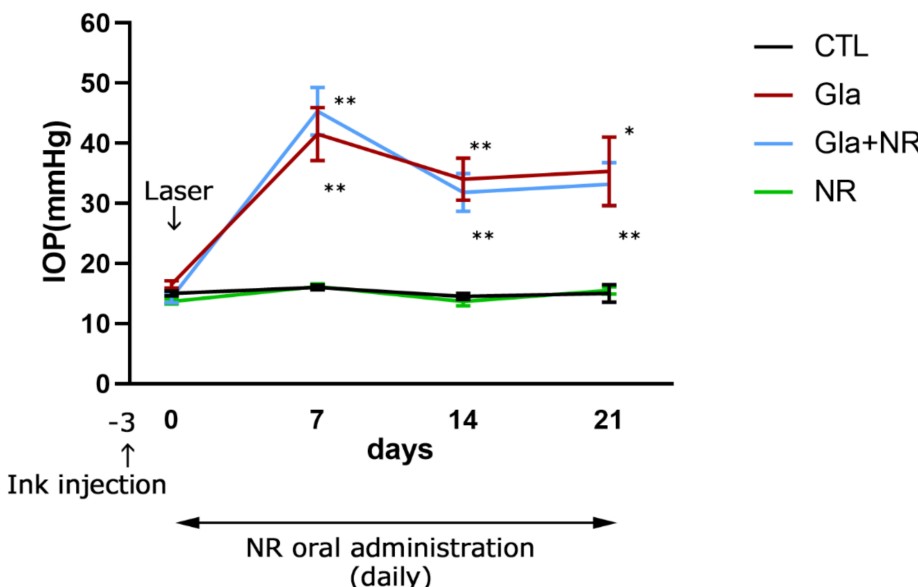

**Figure 1.** Time course of IOP changes in the control (*n* = 4), experimental glaucoma (*n* = 6), glaucoma + NR (*n* = 6), and NR alone (*n* = 6) groups. Significant differences in IOP values were observed in laser-treated groups compared with the control group at 7, 14, and 21 days (*p* * < 0.05 vs. CTL; *p* ** < 0.005 vs. CTL).

*3.2. NAD Measurement*

NAD is one of the most important and abundant molecules in the human body, and $NAD^+$ levels decline gradually with age, leading to increased disease susceptibility [24]. This decline in the $NAD^+$ level with age is observed in the retina [8]. Supplementation to increase $NAD^+$ levels includes nicotinic acid, niacin, nicotinamide, and NR. Two previous studies demonstrated that intraperitoneal injection of NR (1000 mg/kg daily for 5 days and for 2 days) increased retinal $NAD^+$ levels in mice [25,26]. In the current study, we examined the effect of daily oral NR administration on retinal $NAD^+$ levels in rats. NAD assay results showed a significant increase in $NAD^+$ levels in the retinas of rats administered with NR compared with control retinas (Figure 2).

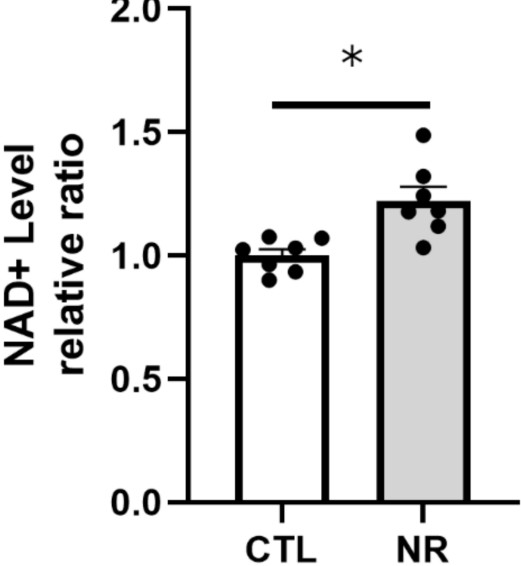

**Figure 2.** NAD assay in retinas after 1 week of daily oral NR administration. (CTL: *n* = 7, NR: *n* = 7) (*p* * < 0.05).

### 3.3. Effects of NR Administration on Optic Nerve Axonal Degeneration after IOP Elevation

Histological findings showed substantial degenerative changes and apparent axon loss after IOP elevation (Figure 3B) compared with the controls (Figure 3A). Quantitative analysis showed approximately 50% axon loss ($p < 0.0001$ vs. CTL; Figure 3D). On the other hand, oral NR administration in the ocular hypertension group showed apparent protective effects (Figure 3C) compared with the ocular hypertension group without NR administration (Figure 3B). Quantitative analysis showed substantial protective effects against ocular hypertensive axon loss ($p < 0.0001$; Figure 3D).

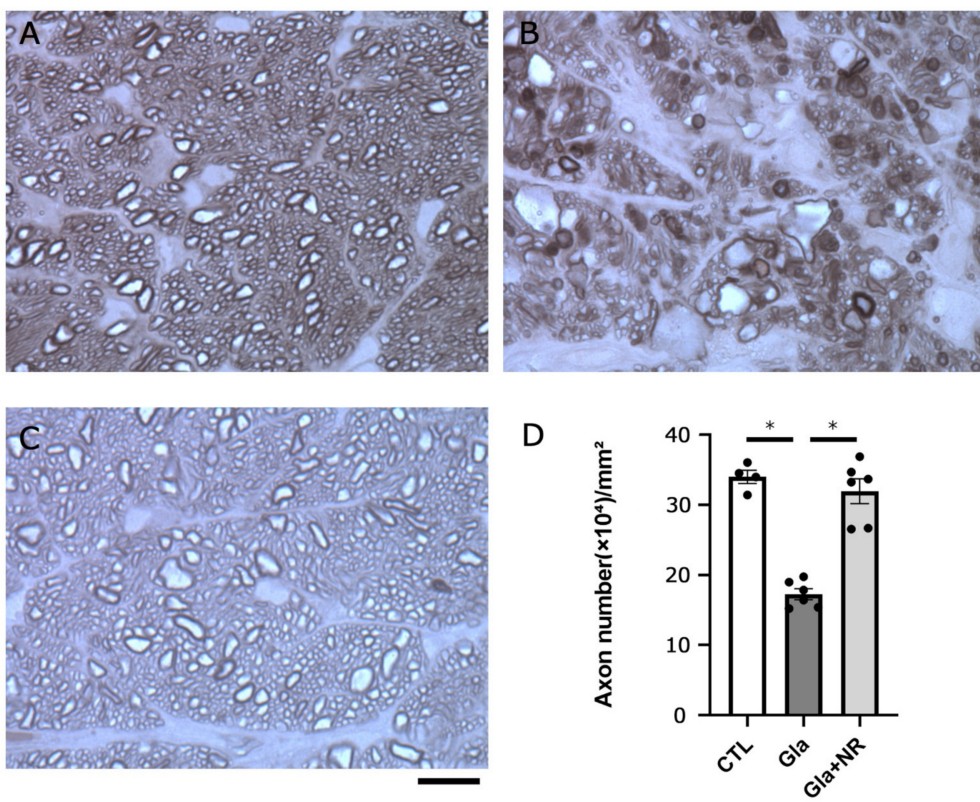

**Figure 3.** Oral NR administration prevented optic nerve axon loss in experimental glaucoma. Optic nerve cross sections after 21 days in the (**A**) control, (**B**) experimental glaucoma, and (**C**) glaucoma + NR groups. Scale bar = 10 μm (**A–C**). (**D**) Effects of NR on axon numbers in the optic nerve. (CTL: *n* = 4, Gla: *n* = 6, Gla + NR: *n* = 6) ($p$ * < 0.0001).

### 3.4. Effects of NR Administration on Retinal Axon Degeneration after IOP Elevation

Histological findings showed notable neurofilament-positive nerve fiber losses in the retina after IOP elevation (Figure 4B,E) compared with the controls (Figure 4A,D). Quantitative analysis of pixel values showed that there were significant decreases in nerve fibers in the ocular hypertension group compared with the controls at the center and periphery ($p = 0.0206$ vs. CTL, $p = 0.0021$ vs. CTL, center and periphery, respectively; Figure 4). On the other hand, oral NR administration with the ocular hypertension group showed noticeable protective effects (Figure 4C,F) compared with the ocular hypertension group not administered NR (Figure 4B,E). Quantitative analysis of pixel values showed significant protective effects of NR against ocular hypertensive neuronal damage at the center and periphery ($p = 0.0388$ and $p = 0.0031$ vs. no NR administration, center and periphery, respectively; Figure 4G,H).

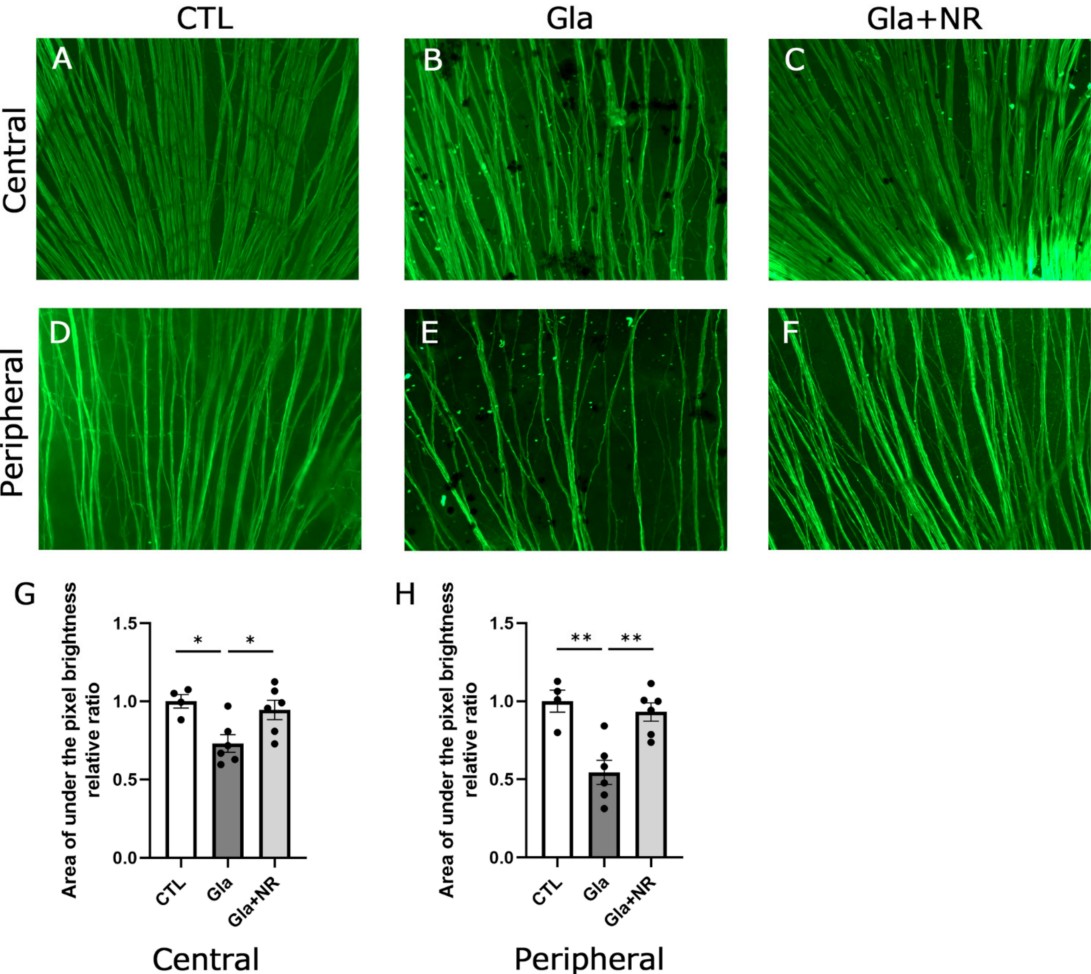

**Figure 4.** Oral NR administration prevented RGC fiber loss in experimental glaucoma. Flat-mounted neurofilament-positive fibers after 21 days in the (**A**) control, (**B**) experimental glaucoma, and (**C**) glaucoma + NR groups in the central area. Flat-mounted neurofilament-positive fibers after 21 days in the (**D**) control, (**E**) experimental glaucoma, and (**F**) glaucoma + NR groups in the peripheral area. Scale bar = 100 μm. (**G**) Quantitative analysis of pixel values in the central area. (**H**) Quantitative analysis of pixel values in the peripheral area. (CTL: $n = 4$, Gla: $n = 6$, Gla + NR: $n = 6$) ($p$ * $< 0.05$; $p$ ** $< 0.005$).

### 3.5. Effects of NR Administration on Optic Nerve p-AMPK Protein Levels

Our ocular hypertension model displays apparent axon loss 3 weeks after IOP elevation [21]. Thus, the molecular events at 1 week, before axon loss becomes apparent, are important for elucidating the mechanism of axonal damage. Therefore, we investigated the immunoblot of p-AMPK in optic nerve samples 1 week after IOP elevation. The current study found no significant difference in p-AMPK levels between the control and ocular hypertension groups (Figure 5). Interestingly, oral NR administration in the ocular hypertension group showed a pronounced increment in p-AMPK levels compared with the ocular hypertension group that did not receive NR (Figure 5). In addition, oral NR significantly increased the p-AMPK level compared with the control group (Figure 5), indicating that oral NR augments the optic nerve p-AMPK level both with and without ocular hypertension induction. Next, we examined how long AMPK phosphorylation lasts after one-time NR administration. We found that there was a significant increase in p-AMPK levels at 24 h, but not at 6 and 48 h after oral NR administration (Figure 6), implying that the daily oral NR administration may maintain incremental p-AMPK levels.

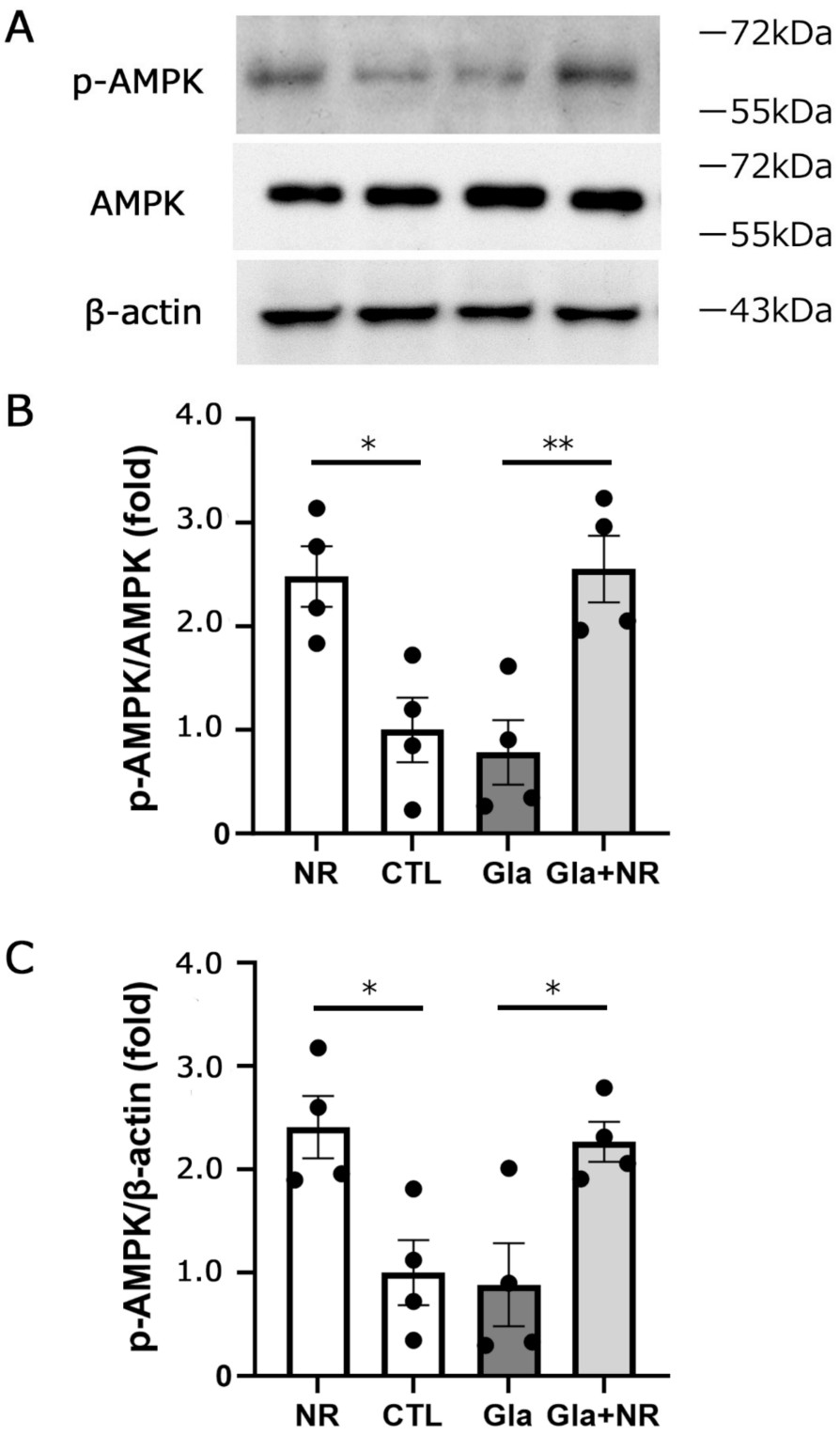

**Figure 5.** Oral NR administration augmented p-AMPK levels in the optic nerve. Immunoblotting in samples from optic nerves 1 week after laser irradiation (**A**). Effects of experimental glaucoma, glaucoma + NR, or NR alone on p-AMPK protein level. Data are expressed as the p-AMPK/AMPK ratio (**B**) and p-AMPK/β-actin ratio (**C**) (CTL: *n* = 4, Gla: *n* = 4, Gla + NR: *n* = 4, NR alone: *n* = 4) (*p* * < 0.05; *p* ** < 0.01).

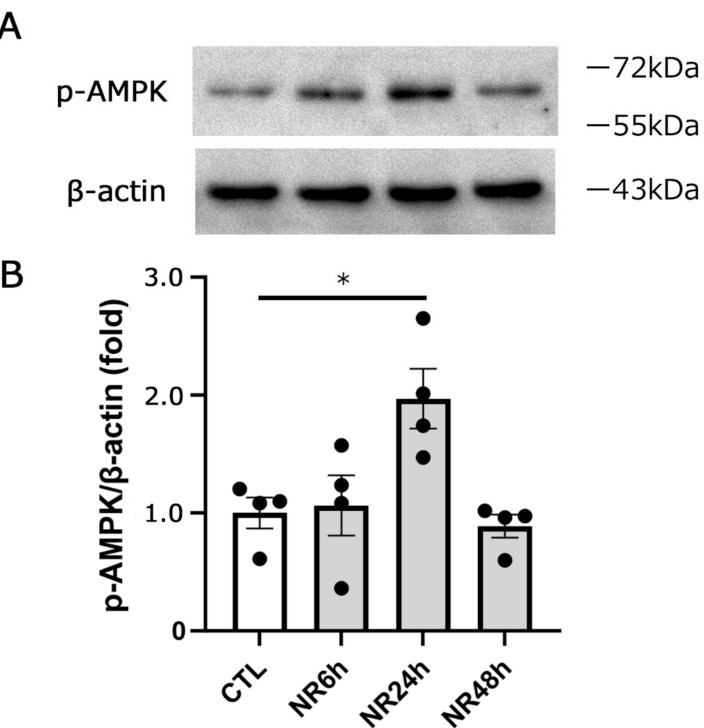

**Figure 6.** One-time oral NR administration transiently augmented p-AMPK levels in the optic nerve. Immunoblotting in samples from optic nerves in the control, 6, 24, and 48 hrs after NR administration (**A**). Data are expressed as the p-AMPK/β-actin ratio (**B**) (CTL: $n = 4$, 6 h: $n = 4$, 24 h: $n = 4$, 48 h: $n = 4$) ($p * < 0.05$).

### 3.6. Localization of p-AMPK in the Optic Nerve

In agreement with our recently reported results [27], the p-AMPK immunoreactive pattern was similar to that of neurofilament immunoreactivity (Figure 7). The immunoreactivity of p-AMPK was modest, but some immunopositive fibers were colocalized with immunoreactive neurofilaments in the control group (Figure 7). After oral NR administration, many p-AMPK-immunopositive fibers were colocalized in the optic nerve with neurofilament immunoreactivity (Figure 7).

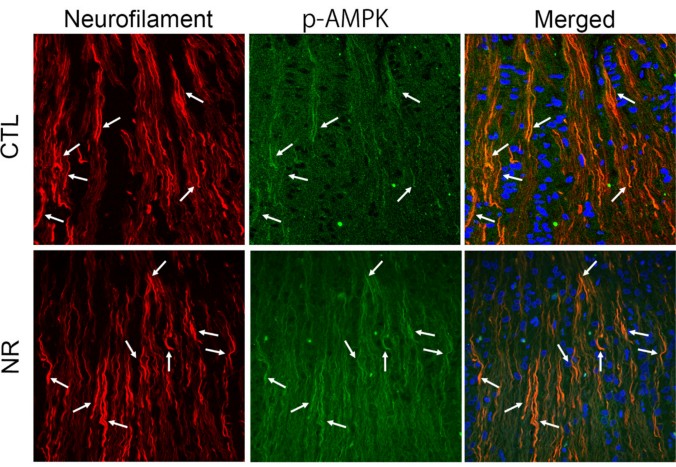

**Figure 7.** Oral NR administration enhanced p-AMPK immunoreactivity in the optic nerve. Immunohistochemistry showed that p-AMPK immunoreactivity was modest and colocalized with neurofilament-positive fibers in the control group. Enhanced p-AMPK immunoreactivity was observed with neurofilament-positive fibers 1 week after NR administration. Arrows indicate colocalization. Scale bar = 50 μm.

## 4. Discussion

The present study showed that chronic IOP elevation caused retinal nerve fiber loss as shown by neurofilament staining. These elevated IOP-induced changes in neurofilament-positive fibers were more prominent in the peripheral than in the central region. Although the model was different, it was reported that acute IOP elevation (60 mmHg for 2 h) caused retinal neurofilament-positive fiber loss, which was also more prominent in the peripheral than in the central region [28]. This may be due to the small number of nerve fibers in the peripheral region. Some studies using the DBA/2J glaucoma mouse model demonstrated fan-shaped RGC axon loss in the retina, but these neurofilament-positive fibers are fewer in the peripheral region [29,30]. Importantly, the latter studies reported that axonal atrophy preceded RGC death [29,30]. However, caution should be needed because other groups showed that axotomy-induced RGC death precedes the loss of intraretinal axons [31]. That study used an antibody against phosphorylated neurofilament subunit H in the axotomy model, but our current study used an antibody against non-phosphorylated neurofilament subunit L in the ocular hypertensive model. These differences may affect the interpretation of RGC death and axon loss.

The present study revealed that oral NR administration (1000 mg/kg daily) exerted substantial axonal protection in ocular hypertension optic nerve damage. We also found apparent protective effects on retinal nerve fibers in this model. In agreement with these findings, a recent study demonstrated that intraperitoneal injection of NR (1000 mg/kg 3 times weekly) prevented RGC loss in microbead-injection-induced glaucoma model mice [25]. That study also demonstrated that intraperitoneal injection of NR (1000 mg/kg 3 times weekly) prevented RGC loss in optic nerve crush model mice [25]. Thus, although it is likely that systemic administration of NR protects both RGCs and their axons, to our knowledge, this is the first study demonstrating axonal protection of NR against glaucomatous damage in both the optic nerve and retina. In other neuronal axons, some studies found beneficial effects of NR in distinct axonal damage models. For example, a previous study demonstrated that NR attenuated axonal degeneration in dorsal root ganglia neurons [32]. Moreover, another study showed that NR suppressed excitotoxicity-mediated axonal degeneration in cortical neurons [33]. A very recent study found that NR can prevent tri-ortho-cresyl phosphate-induced axonal degeneration of the dorsal root ganglia neurons [34]. Since axonal degeneration precedes RGC body death in glaucoma, oral NR administration may help to inhibit further axonal degeneration, thereby preventing additional RGC death.

It is interesting to note the difference in IOP changes with NR administration in the above mouse glaucoma model and our rat glaucoma model. They demonstrated that intraperitoneal injection of NR unexpectedly prolonged IOP elevation in microbead-injected mice [25], whereas our current study showed that oral NR had no effect on IOP in the laser-induced glaucoma rat model, although both studies found that NR had no effect on IOP in the control groups. One hypothesis posits that different glaucoma models and different methods of NR administration may affect the types of IOP changes.

Exogenous NR can be converted to $NAD^+$ via nicotinamide riboside kinase 1 (NRK1) in mammalian tissues [35]. Expression of NRK1 was observed in the brain, heart, kidney, liver, muscle [35], retina, and optic nerve [36]. Intraperitoneal injection of NR significantly upregulated $NAD^+$ levels in the muscle, liver, and brain [35]. On the other hand, oral NR is more available than oral nicotinamide for increasing $NAD^+$ serum levels and hepatic $NAD^+$ metabolism [1]. In the present study, we found that oral administration of NR significantly increased $NAD^+$ levels in the retina. Thus, it is assumed the metabolism of the latter occurs more systemically, transporting $NAD^+$ to the bloodstream and then reaching the retina.

Our previous study demonstrated that intravitreal injection of NR attenuated axonal degeneration via the SIRT1-autophagy pathway in the TNF-induced optic nerve damage model [36]. In this TNF-induced optic nerve degeneration model, we recently found that netarsudil, a ROCK inhibitor, upregulated the p-AMPK protein level in the optic nerve and ameliorated axonal loss [27]. In the current study, oral NR administration clearly increased

p-AMPK levels in the optic nerve in glaucomatous as well as control eyes. NR can be converted into nicotinamide mononucleotide (NMN) by NRK1, while nicotinamide can be converted into NMN by nicotinamide phosphoribosyltransferase (Nampt). NMN can then be converted into $NAD^+$ by nicotinamide mononucleotide adenylyltrasferase (Nmnat). A recent study showed reductions in Nampt, Nmnat1, and Nmnat2 in RGCs in glaucoma patients [37], implying that not only the $NAD^+$ level but also the NAD salvage pathway can be reduced in glaucoma. Therefore, NR, NRK1, Nampt, NMN, and Nmnat are upstream contributors to the $NAD^+$ synthesis pathway. Although a direct association between NR and AMPK has not been explored in neurons, it is noteworthy that Nampt ameliorated acute ischemic stroke during neuronal injury by upregulated AMPK [38]. We recently found that p-AMPK exists in optic nerve axons [27], and the present study found that oral NR administration enhanced the axonal immunoreactivity of p-AMPK. This is consistent with the current immunoblotting findings, implying that oral NR administration upregulates p-AMPK in optic nerve axons. Because one-time oral NR administration leads to p-AMPK elevation at 24 h, it is reasonable to speculate that daily oral NR administration leads to p-AMPK elevation during oral administration. One possible underlying mechanism of the upregulation of p-AMPK and protection from ocular hypertension is that upregulation of p-AMPK may lead to autophagy activation, because AMPK activator A769662 exerted axonal protection with autophagy activation in TNF-induced optic nerve damage [27]. In other neurons, NR upregulated the expression of key proteins of lipid droplet dynamics, PLIN1 and CIDEC, in the cochlea [39]. It is interesting to note a very recent study showing that NR was identified as a SIRT5-selective activator [40]. Thus, further studies will be necessary to fully understand the molecular mechanism implicated in NR neuroprotection.

In conclusion, it is suggested that oral NR administration exerts protection against glaucomatous RGC axonal degeneration with possible upregulation of p-AMPK.

**Supplementary Materials:** The following supporting information can be downloaded at: https://www.mdpi.com/article/10.3390/cimb45090449/s1. Supplementary Figure S1. The quantitative pixel brightness values were expressed as the area of under curve which means total sum of pixel value in the analysis line length. Red analysis line = 500 μm. Supplementary Figure S2. Time course of IOP changes in the sham (*n* = 6) and experimental glaucoma (*n* = 6) groups. Although IOP tended to be higher in the laser group compared to the no-laser group (sham), statistical significance was observed in only day 1 (*p* * < 0.05 vs. sham).

**Author Contributions:** I.A. performed the intracameral injections, laser irradiation, and immunohistochemistry and immunoblot analysis; measured IOP; and wrote the manuscript. N.F. performed the intracameral injections, laser irradiation, and the procedure for flat-mount immunostaining; and measured IOP. C.T. and N.T. performed the protein assays and prepared several samples for immunoblotting and immunohistochemistry. K.S. performed axon counting and immunoblot analysis and interpreted data. R.S. prepared for the flat-mount immunostaining procedure. T.J. prepared several samples for immunohistochemistry. M.O. prepared several samples for immunoblotting. Y.K. designed the entire study, performed laser irradiation, enucleated eyes, and wrote the manuscript. All authors have read and agreed to the published version of the manuscript.

**Funding:** This study was supported by Grants-in-Aid from KAKENHI (22K09843 for Y.K.; 23K15944 for K.S.; 22K20981 for N.F.).

**Institutional Review Board Statement:** All studies were approved by the Ethics Committee of the Institute of Experimental Animals, St. Marianna University School of Medicine (No. 2108012, No. 2208007).

**Informed Consent Statement:** Not applicable.

**Data Availability Statement:** The data presented in this study are available in the submitted article.

**Conflicts of Interest:** The authors declare that they have no competing interests.

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
