# Peer review of "Axonal Protection by Oral Nicotinamide Riboside Treatment with Upregulated AMPK Phosphorylation in a Rat Glaucomatous Degeneration Model"

_cimb, doi:10.3390/cimb45090449_

Round 1
Reviewer 1 Report
The main purpose of this study is to evaluate the neuroprotective effect of NR in a rat model of glaucoma induced by laser irradiation. NR is a precursor of nicotinamide adenine dinucleotide and it has become the focus of study as a promising treatment for degenerative diseases. Recent studies have also shown the beneficial effect of NR treatment in glaucoma and therefore, this topic is of great interest.
The main finding of this manuscript is that NR induces neuroprotection of axons through increased p-AMPK phosphorylation in the optic nerve at 7 days. In my opinion, this result is novel and may contribute to the knowledge of NR, but it is the only novel result and the study is not sufficient for publication for the following reasons:
First, the authors used the laser irradiation model to induce OHT and retinas were analysed after 7, 14 and 21 days. In this model, IOP reaches 40 mmHg after 7 days, which makes this model very acute. It is not clear why the authors chose 7 days as the first time of analysis. It would be important to show when IOP is first significantly higher (e.g. 1, 3, 5 days).
This is also important to understand the molecular mechanisms involved in axonal and RGC degeneration. The activation of signalling pathways in glaucoma depends on the experimental model used and the time point analysed. In general, the phosphorylation or activation of proteins related to this type of insult is fast and it usually occurs earlier than 7 days. It would be very interesting to analyse when pAMPK is first activated and how long this phosphorylation lasts.
Moreover, although previous studies have described RGC survival in OHT, variability between experimental models should be taken into account. It would be very helpful to correlate the density of surviving RGCs (Brn3a+ or RBPMS+) with AMPK signalling pathway to confirm that, when AMPK is not phosphorilated anymore, RGCs die faster, for instance.
Finally, pAMPK immunoreactivity in RGC axons is a novel finding but RGC axonal degeneration after NR treatment has been described before (PMID: 33932867). In addition, the results of axonal degeneration showed here may be extended. For instance, the authors showed that the density of axons was lower in the periphery than in the centre of the retina. When analyzing the optic nerve, it would be interesing to know if this sectorial damage is similar. Accumulation of neurofilament is also observed in RGCs (figure 4), as it has been described as a sign of degeneration (PMID: 29452106). It would be interesting to explore these findings further.
Finally, the discussion section is insufficient and the number of references must be extended to fully understand the molecular mechanisms implicated in NR neuroprotection.
The manuscript is well-written.
Reviewer 2 Report
Original submission
1.1 Recommendation
Minor Revision
2. Comments to Author:
Titel: Axonal protection by oral nicotinamide riboside treatment with 2 upregulated AMPK in a rat glaucomatous degeneration model
Ibuki Arizono, Naoki Fujita, Chihiro Tsukahara, Kana Sase, Reio Sekine, Tatsuya Jujo, Mizuki Otsubo, Naoto Tokuda and Yasushi Kitaoka
Overview and general recommendation:
In their study the authors show that oral administration of nicotinamide riboside protects optic nerve axons in a rat experimental glaucoma model and that NR increases p-AMPK levels in the optic nerve.
While big parts of the study are focusing on showing that NR administration protects ON axons and retinal fibers in their experimental glaucoma model, which has been shown before for other glaucoma models multiple times, only a small part of the results, with very low n numbers, presents new findings.
Overall this study should be published after revision only.
2.1 Comments:
- Figure 5: Regarding the small sample size and partially high standard deviation choice of a ANOVA for statistical testing might be wrong.
Could the authors please provide all western blot images and quantifications? Furthermore, quality of AMPK western blot is not that good, bands are likely hard to quantify.
- Phrasing “Upregulated AMPK”: the authors state that the protective effect is accompanied by upregulated AMPK, while the study shows upregulated phosphorylation of AMPK, changes in unphosphorylated AMPK were not quantified, but it the discussion the statement is made that “AMPK immunoreactivities were unchanged between the control and NR administration groups evaluated by current immunoblotting as well as immunohistochemical analysis” (L 329-331)
- Material and Methods, Ocular hypertension model: please describe in more detail what experimental groups and control groups are. Is ocular hypertension induced in both eyes of the rats or is one eye left untreated as intra-animal control? Do control animals also get injections of India ink etc.? Furthermore, the source for the model does not lead to a publication with a clear description of the model
- 3.2 NAD measurements “one previous study demonstrated that intraperitoneal injection of NR (1000 mg/kg daily for 5 days) increased retinal NAD+ level in mice” à there are multiple studies showing this effect, e.g. Chen et al., 2020 and Zhang et al., 2020
- Could the authors please explain why NFH staining of retinae was chosen to show protection of RGC cells and not more common methods like RBPMS or BRN3a staining with quantification of RGC somata
Author Response
We very much appreciate your kind consideration and the helpful comments. All the issues that were brought up were taken into account in the revised manuscript. We believe that these helpful comments improved the manuscript greatly. Our responses to the comments are given below.
Titel: Axonal protection by oral nicotinamide riboside treatment with 2 upregulated AMPK in a rat glaucomatous degeneration model
Ibuki Arizono, Naoki Fujita, Chihiro Tsukahara, Kana Sase, Reio Sekine, Tatsuya Jujo, Mizuki Otsubo, Naoto Tokuda and Yasushi Kitaoka
Overview and general recommendation:
In their study the authors show that oral administration of nicotinamide riboside protects optic nerve axons in a rat experimental glaucoma model and that NR increases p-AMPK levels in the optic nerve.
While big parts of the study are focusing on showing that NR administration protects ON axons and retinal fibers in their experimental glaucoma model, which has been shown before for other glaucoma models multiple times, only a small part of the results, with very low n numbers, presents new findings.
Overall this study should be published after revision only.
2.1 Comments:
- Figure 5: Regarding the small sample size and partially high standard deviation choice of a ANOVA for statistical testing might be wrong.
Could the authors please provide all western blot images and quantifications? Furthermore, quality of AMPK western blot is not that good, bands are likely hard to quantify.
Response
Because rat optic nerves are very small, we used two optic nerves from two rats as one sample. Therefore, increase of sample size means twice rat numbers. As suggested, we uploaded all western blot images.
Quantification data of p-AMPK/b-actin are ,
NR(1.957789, 3.176885, 2.600915, 1.898869)
CTL(0.723063, 1.121326, 1.81188, 0.34373),
Gla(0.899405, 0.32723, 0.293405, 2.012818)
Gla+NR(1.907457, 2.059577, 2.315171, 2.790535).
Quantification data of p-AMPK/AMPK are ,
NR(2.17961978, 3.137753837, 2.769851033, 1.835035141)
CTL(0.84987981, 1.198369357, 1.722497195, 0.229253637),
Gla(0.905958355, 0.264228301, 0.346304676, 1.616598237)
Gla+NR(2.052759967, 1.960509343, 3.236046261, 2.961493407).
Also, as suggested, we performed additional AMPK western blot with a different company antibody. We replaced with the better photo and revised the graph. We revised the figure 5 and Methods accordingly.
- Phrasing “Upregulated AMPK”: the authors state that the protective effect is accompanied by upregulated AMPK, while the study shows upregulated phosphorylation of AMPK, changes in unphosphorylated AMPK were not quantified, but it the discussion the statement is made that “AMPK immunoreactivities were unchanged between the control and NR administration groups evaluated by current immunoblotting as well as immunohistochemical analysis” (L 329-331)
Response
We apologize our mistake. We do not show the AMPK immunoreactivity, therefore we deleted the phrase “Moreover, we observed that AMPK faintly exists in optic nerve axon. These AMPK immunoreactivities were unchanged between the control and NR administration groups evaluated by current immunoblotting as well as immunohistochemical analysis”.
We appreciate this important pointed out. Also, as suggested, we revised the title, abstract, and discussion as below.
“Axonal protection by oral nicotinamide riboside treatment with upregulated AMPK phosphorylation in a rat glaucomatous degeneration model”
“with the possible upregulation of p-AMPK.”
“In conclusion, it is suggested that oral NR administration exerts protection against glaucomatous RGC axonal degeneration with possible upregulation of p-AMPK.”
- Material and Methods, Ocular hypertension model: please describe in more detail what experimental groups and control groups are. Is ocular hypertension induced in both eyes of the rats or is one eye left untreated as intra-animal control? Do control animals also get injections of India ink etc.? Furthermore, the source for the model does not lead to a publication with a clear description of the model
Response
We described in more detail. India ink was injected to the right eye. The contralateral left eye without India ink was used as control or NR alone group. Therefore, one eye left is untreated as intra-animal control. Also, we performed additional experiments for sham study which control gets injection of India ink. We revised the Methods and Results as below and added supplemental figure 2.
“The contralateral left eye without India ink was used as control or NR alone group. For sham study, we injected India ink intracamerally to the right eye and compare the IOP between India ink and India ink with laser irradiation.”
“For sham study, IOP was measured baseline, 1, 3, and 5 days after laser irradiation.”
“In the sham study, there was no significant difference in IOP at baseline between two groups (supplemental figure 2). Although IOP tended to be higher in the laser group compared to no laser group, statistical significance was observed in only day 1 (supplemental figure 2).”
- 3.2 NAD measurements “one previous study demonstrated that intraperitoneal injection of NR (1000 mg/kg daily for 5 days) increased retinal NAD+ level in mice” à there are multiple studies showing this effect, e.g. Chen et al., 2020 and Zhang et al., 2020
Response
Thank you so much for this pointed out. Perhaps, Chen et al., 2020 is PMID 33373320. This study performed intraperitoneal injection of nicotinamide mononucleotide (NMN) (500 mg/kg daily for 5 days). Because the current study used NR, we cited the latter study by Zhang et al., 2020, PMID 32852543. As pointed out, the latter study performed intraperitoneal injection of NR (1000 mg/kg daily for 2 days). We revised the Result as below,
“Two previous studies demonstrated that intraperitoneal injection of NR (1000 mg/kg daily for 5 days and for 2 days) increased retinal NAD+ level in mice [25, 26].”
- Could the authors please explain why NFH staining of retinae was chosen to show protection of RGC cells and not more common methods like RBPMS or BRN3a staining with quantification of RGC somata.
Response
Our model may cause RGC death 5 weeks after IOP elevation (PMID 20950337). This time, we focused on axonal degeneration which can be seen 3 weeks after IOP elevation. If we would assess RGC death in our model, we needed to perform daily oral NR administration for 5 weeks. It is very tough to perform daily oral NR administration for 5 weeks. Actually, it was tough to perform daily oral NR administration for 3 weeks. So, we appreciate it if the reviewers would understand our situation.